

# Short-term wind power forecasting through stacked and bi directional LSTM techniques

Mehmood Ali Khan[1,*], Iftikhar Ahmed Khan[2], Sajid Shah[3], Mohammed EL-Affendi[3] and Waqas Jadoon[2,*]

[1] Computer Science, Virtual University, Islamabad, Federal, Pakistan
[2] Computer Science, COMSATS University Islamabad, Abbottabad, KpK, Pakistan
[3] EIAS Data Science and Blockchain Lab, College of Computer and Information Sciences, Prince Sultan University, Riyadh, Saudi Arabia
[*] These authors contributed equally to this work.

## ABSTRACT

**Background**. Computational intelligence (CI) based prediction models increase the efficient and effective utilization of resources for wind prediction. However, the traditional recurrent neural networks (RNN) are difficult to train on data having long-term temporal dependencies, thus susceptible to an inherent problem of vanishing gradient. This work proposed a method based on an advanced version of RNN known as long short-term memory (LSTM) architecture, which updates recurrent weights to overcome the vanishing gradient problem. This, in turn, improves training performance.

**Methods**. The RNN model is developed based on stack LSTM and bidirectional LSTM. The parameters like mean absolute error (MAE), standard deviation error (SDE), and root mean squared error (RMSE) are utilized as performance measures for comparison with recent state-of-the-art techniques.

**Results**. Results showed that the proposed technique outperformed the existing techniques in terms of RMSE and MAE against all the used wind farm datasets. Whereas, a reduction in SDE is observed for larger wind farm datasets. The proposed RNN approach performed better than the existing models despite fewer parameters. In addition, the approach requires minimum processing power to achieve compatible results.

# INTRODUCTION

There is an ongoing demand for renewable energy sources to address global warming, fossil fuel depletion, and electricity demand. Wind power is a popular option because it is eco-friendly, clean, viable, and cheap. Furthermore, there is a huge global potential of 600 gigawatts of wind energy (*Nazir et al., 2020*). Wind power prediction is paramount in optimizing resource utilization within the energy sector. Accurate predictions play a pivotal role in steering away from traditional energy sources and embracing a more

Corresponding author
Iftikhar Ahmed Khan,
mailtoiftikhar@gmail.com

sustainable future. Failing to grasp the practical ramifications of inaccurate wind power predictions can result in inefficient energy distribution, elevated operational costs, and adverse environmental impacts.

This underscores the necessity for sophisticated models such as long short-term memory (LSTM) in tackling the complexities inherent in wind power forecasting. However, it's crucial to bridge the gap between the importance of accurate predictions and the utilization of advanced models. In this respect, many non-machine learning (nML) as well as machine learning (ML) techniques have been developed to predict wind power and speed using weather data and historical records such as *Jamii et al. (2022)*, *Mujeeb et al. (2019)* and *Magadum et al. (2023)*. ML techniques such as neural networks and LSTM can estimate wind speed better than archaic numerical weather prediction methods (*Zhang et al., 2022*; *Peng et al., 2021*; *Peng et al., 2020*; *Kumar & Yadav, 2024*; *Wang et al., 2024*).

Conventional non-machine learning autoregressive models capture temporal dependencies in time series data, can handle sequential information, and learn patterns from historical observations just like LSTM. However, autoregressive models assume fixed gaps compared to LSTM which can capture non-linear dependencies. The LSTM can adapt to variable gaps. LSTM can also address vanishing gradient problems and model long-term dependencies more effectively than auto-regressor models. Furthermore, both nML models such as moving averages and ML models *i.e.,* LSTMs can be used to smooth out variations in time series data as well as both can handle noise and highlighting trends. However, moving averages may not capture non-linear trends and long-term dependencies. LSTMs are also more adaptive as they can adjust weights dynamically and therefore are more suitable for handling variations in wind data. Therefore, the following ML studies highlight the wind's speed and power prediction.

*Hasheminejad & Fekri (2009)* used an adaptive neural network, *Han, Yang & Liu (2010)* used an improved neural network with wind direction patterns, and *Su et al. (2010)* used feedforward backpropagation and Levenberg Marquardt algorithms. *Mohammadi et al. (2015)* used extreme learning machines (ELM) and compared them with artificial neural network (ANN), support vector machine (SVM), and genetic programming (GP) models. They found that ELM was accurate and precise as compared to other algorithms. *Najeebullah et al. (2015)* used a hybrid approach with SVR and PSO. *Treiber, Heinermann & Kramer (2016)* used regression and K-nearest neighbor. *Deepa Lakshmi & Sujatha (2016)* used ANN as a tool for continuous monitoring of wind speed.

In a similar context, *Qureshi et al. (2017)* used an ensemble of deep sparse auto-encoders as base regressors and a deep belief network as a meta-regressor. *Yue et al. (2017)* used ensemble empirical mode decomposition, sample entropy, and least-square supporting vector machines optimized by particle swarm optimization (PSO). *Yadav & Sahu (2017)* surveyed different AI-based models and activation functions for wind speed forecasting. *Zameer et al. (2017)* used a combination of genetic programming and artificial neural network (GPeANN) ensemble technique with previous power/forecast values. *Zhang et al. (2018)* used Lorenz disturbance, principal component analysis, and improved particle swarm optimization (LD-PCA-IPSO-BPNN) model. *Zucatelli et al. (2019)* used an

ANN-based model with multiple layers of the Levenberg–Marquardt backpropagation AI technique to forecast wind speed at different time intervals.

In recent years, interest in deep artificial neural networks (D-ANN) has increased due to improvements in the prediction of both small-range and long-range reliance in data sequences as compared to shallow neural networks. D-ANN are better at predicting data sequences than shallow neural networks. LSTM is a D-ANN that remembers and reuses past weights and thus can handle the deep nonlinear behavior of wind speed. LSTM has been used in engineering areas such as signal classification (*Hamza et al., 2022*) and software radio (*Su et al., 2010*) *etc*. Literature shows that LSTM and its variants have been used for wind power prediction. *Yu et al. (2019)* used LSTM with a prediction window and sequential sequence to predict wind power for different time horizons. LSTM performed better than other models for longer predictions. *Araya, Valle & Allende (2019)* used an LSTM-based multi-scale model (LSTM-Ms) with Feed Forward networks for temporal scale feature extraction which outperformed the persistence model and standard LSTM. LSTM is also found to be better than SVM and RNN for wind speed forecasting (*Gangwar, Bali & Kumar, 2020*; *Lipton, Berkowitz & Elkan, 2015*).

With the advancements of new techniques in the domain of ML & CI, researchers have been able to find more advanced and novel techniques based on wind and weather forecast datasets to mitigate power management issues. This is achieved either by short-term or long-term wind power predictions. Since the ML techniques are primarily based on a dataset, no single technique can be considered best for all wind data collected at different wind farms located at different geographical locations. Therefore, there is a need for constant improvement of either datasets or techniques for better and more effective models depending on the dataset available.

A comparison of LSTM with regression, random forest, and gradient boosting techniques was done by *Dong, Sheng & Yang (2019)*. They claimed to provide a novel predicting framework with an effective forecasting map adapted to different prediction horizons. However, short-term wind speed prediction using LSTMs', or their variants is not addressed in any of the above-mentioned studies. Recently *Ibrahim et al. (2020)* compared ANN, CNN, LSTM, a hybrid convolutional LSTM (convLSTM), and SVM for short-term wind speed prediction. ConvLSTM gave better results both in terms of performance and accuracy. However, the research compared LSTM with ANN, CNN, and convLSTM which are legacy techniques. This research aims to compare LSTM against more novel and hybrid techniques. Moreover, *Ibrahim et al. (2020)* work is specifically on wind speed forecast. The target of this research is focused on wind speed as well as wind power in the feature set.

*Qureshi et al. (2017)* and *Zameer et al. (2017)* presented state-of-the-art techniques that used a combination of an ensemble of neural networks with genetic programming and transfer learning to enhance prediction accuracy. Even though these techniques provided promising results, their implementation of feature engineering and the combination of different techniques make these models more complex and resource intensive. Our research is more focused on finding a solution that is simpler, more efficient, and should have acceptable accuracy as well. To achieve this, we reduced the feature set by 70% and

replaced complex ensembles of ANN and DNN with the simple implementation of LSTM. For the efficiency assessment of the proposed solution, the results are compared with the existing wind speed prediction techniques presented by *Zameer et al. (2017)* and *Qureshi et al. (2017)*. The present work uses LSTM variants, Stacked LSTM, and bidirectional LSTM for wind power prediction because of their simplicity and their successful implementation in other relevant problems. It is pertinent to note that some latest related work such as (*Xie et al., 2021*; *Liang et al., 2021*; *Prema et al., 2019*) used the prediction error values of wind speed and wind power for the next hour). Therefore, this research has also utilized an algorithm for the next hour prediction of wind speed and power.

The rest of the article is organized as follows. 'Proposed Technique for Short Term Wind Power Prediction' discusses a proposed methodology for short-term wind power prediction. 'Results and Discussion' describes the results of the applied methodologies. Finally, 'Conclusions, Limitations and Future Work' concludes the article.

# PROPOSED TECHNIQUE FOR SHORT TERM WIND POWER PREDICTION

In this study, LSTM variants, Stacked LSTM, and bidirectional LSTM are used as the proposed methodology. In the following section, we will discuss the dataset used in this study, LSTM and its variants, and their implementation as the proposed methodology.

## Data collection

The actual data on wind farms is taken from the Kaggle Competition site (*Hongtao, 2012*). The dataset includes power, speed, and directional values for wind besides zonal (u), and meridional (v) components. The dataset is compiled at the wind farm level. The total number of instances available in each wind farm dataset are Wind Farm 1: 16,589, Wind Farm 2: 16,586, Wind Farm 3: 17,010, Wind Farm 4: 15,385, and Wind Farm 5: 16,531, respectively. A mapping of data over hours and wind power is shown in supplementary graphs (Figs. S1–S5). It is noteworthy to mention here that the actual dataset contains data from seven farms while only data from five wind farms is being considered. This is done to ensure a fair comparison between the proposed work and the base literature GPeANN (*Zameer et al., 2017*) and DNN-MRT (*Qureshi et al., 2017*). Each instance of the dataset contains a deterministic forecast, gathered twice a day, and each measured instance has a time-based resolution of 1 h and 2 days ahead. To calculate power, four power values provided by weather predictors, in the last 48 h having a time difference of 12 h are utilized. The dataset also contained temperature information which is not utilized to make a fair comparison with the base articles. The base articles did not utilize temperature for prediction.

## Data pre-processing

The dataset contains weather forecasts for five distinct wind farms along with hourly wind power measurements. During the pre-processing stage, the date and time formats are standardized for observed weather forecasts and power measurements. Metrological forecasts like zonal (u), meridional (v), wind speed (WS), and wind direction (wd) are

used as features to get power measurements. The dataset hour offset format is changed to calculate the exact hour of measuring power output. A simple arithmetic mean is applied to get a single candidate feature set for more than one metrological forecast against a specific power measurement,

Besides weather observations like wind speed and wind direction at specific time intervals, wind power also has a huge dependency on historical weather values. Significant improvement in wind power prediction was observed by *Zameer et al. (2017)* and *Qureshi et al. (2017)* when previous power values were also included in the feature set. To include some specific number of past observations in the feature set, a variable $n$ is defined. This variable $n$ is passed to a function to get a new feature vector that concatenates past n observations. The process is applied to the whole data set. The normalization procedure is considered good practice for inputting data before presenting data to the network (*Hochreiter & Schmidhuber, 1997*). To enhance the accuracy and to make scale-free comparisons, data is scaled down in the range of (0,1). With all the above tuning, the following feature set S(t) as shown in Fig. S6 is finalized.

In the figure, $n$ will be the number of past values that will be included in set S(t). Its range will be from 1 to 24 h in the past. The value of $n$ will be finalized during the training phase based on the performance of the model. Input to the LSTM network is a three-dimensional (3-D) array, these dimensions are the number of training samples to be fed, the number of time steps to be included in a single training example, and the number of features. Data were processed using the Python library *e.g.*, Numerical Python (NumPy), and reshaped as a 3-D array. In our case, each sequence is dealt with independently and the number of the time step is 1. Several features will be as per feature set S(t).

## Long short-term memory networks

LSTM is a specific RNN that is specially designed to deal with data having long-term temporal dependencies. LSTM is an evolved model of classical ANN. Figure S7 shows the architecture as well as the feature set of RNN. As shown in the figure, the output of the previous state is also fed as an input to the subsequent state along with the current input $X_t$. Input to an RNN is the feature vector X and Y will be the output vector containing all the hidden states Rt-1,..., Rt+1.

The recursive nature of RNNs makes them ideal when dealing with sequences. However, *Hochreiter & Schmidhuber (1997)* discovered that the learning process of RNNs becomes drastically slow due to the vanishing gradient problem. This led to the development of LSTM (*Althelaya & Mohammed, 2018*), a special RNN that uses memory units to store contextual information and gated units to supervise the flow of information. In LSTM repeating hidden unit of RNN is replaced with an LSTM cell. A brief overview of LSTM can be viewed in *Hochreiter & Schmidhuber (1997)*.

The thing that differentiates LSTM from standard RNN is its memory cell shown as $C_t$ as highlighted in *Yu et al. (2019)*. The following steps elaborate on the working of LSTM.

- At some time, t, inputs to an LSTM cell are cell state at timestamp t-1 ($C_{t-1}$), the output of the previous hidden state $R_{t-1}$, and the input feature vector $X_t$.

- First, inputs $X_t$ and $R_{t-1}$ are passed to the forget gate (F) along with associated weights that in turn apply the sigmoid activation function on it. Based on the output of the sigmoid function, $F$ decides what information to retain or forget. As sigmoid output values range is between 0 and 1. Values that are closer to 0 will be forgotten and values closer to 1 will remain unchanged. The equation for F will be

$$F = \partial(W^f X_t + W^f R_{t-i}). \tag{1}$$

Next, a two-step process is performed before updating the cell state. First inputs $X_t$, $R_{t-1}$, and associated weights $W^i$ are passed to input gate $I$ which uses the sigmoid function to decide which values to update to the cell state. Values nearest to 1 are likely to be updated. In the second step, utilizing the same input, a candidate vector $G$ is created using *tanh* that holds the new values to be added to the cell state. *Tanh* function returns a value between $-1$ and 1. This will help regularize the network and deal better with the vanishing gradient problem. Now we get the new candidate values through pointwise multiplication of $I$ and $G$. The following equations describe how $I$ and $G$ will be calculated.

$$I = \partial(W^i X_t + W^i R_{t-i}) \tag{2}$$

$$G = tanh(W^g X_t + W^g R_{t-i}). \tag{3}$$

- To update the cell state, the previous cell state $C_{t-1}$ is multiplied by the output of $F$ which is a forget vector. This results in the dropping of cell state values where relevant forget vector values are near 0. Next pointwise addition is applied to the output of the previous step which is $(I \odot G)$, this results in the new cell state $C_t$. Mathematically it can be written as

$$C_t = F \odot C_{t-1} + I \odot G. \tag{4}$$

- Now lastly output will be decided using the output gate ($O$). To calculate the output, feature vector $X_t$, hidden state of previous cell $R_{t-1}$, and relevant weights $W^o$ are used. Based on these inputs $O$ decides the next hidden state $R_t$. Hidden state $R_t$ is the prediction based on previous inputs. First $X_t$ and $R_{t-1}$ are passed to the sigmoid function of $O$. Then the output of $O$ will be pointwise multiplied with the resultant vector of $tanh(C_t)$ where $C_t$ is the new cell state. This multiplication will help decide the relevant information that hidden state $R_t$ should carry. Finally, the hidden state $R_t$ and new cell state $C_t$ will be passed to the next LSTM cell.

$$O = \partial(W^o X_t + W^o R_{t-i}) \tag{5}$$

$$R_t = O \odot tanh(C_t). \tag{6}$$

The final output of an LSTM layer is a vector $Y$ containing all the outputs $(R_{t-1}, \ldots R_{t+1})$ and can be represented as $Y = [R_{t-1}, \ldots R_{t+1}]$.

## Stacked LSTM

Stacked LSTM is the simplest form that enhances the capability of a single LSTM layer network by stacking one or more LSTM layers above it. Figure S8 shows the flow of a stacked LSTM network with a sequence length of 3. In the figure, three recurrent LSTM layers are interconnected where the output of the layer (L-1) *i.e.,* $R_t L - 1$ is treated as input $X_t$ to the subsequent layer (L). Now $X_t$ in the mathematical representation of simple LSTM is replaced with to get the equation for Layer (L) in case of stacked LSTM.

## Bidirectional LSTM

Bidirectional LSTM is another variant of RNN that is found in the literature (*Gao, Xu & Yin, 2024*). It extends the standard unidirectional LSTM to bidirectional thus enabling it to process input sequences in both time directions simultaneously *i.e.,* backward (future to past) or forward (past to future). Here for each time direction, two separate networks are used to train the model. Bidirectional LSTM is applicable where all time steps of the input sequence are known. Results from both layers are then merged to help prediction. Work done by *Althelaya & Mohammed (2018)* on stock market prediction shows that bidirectional LSTM produced better results compared with stacked LSTM.

In bidirectional LSTM the working of the forward layer is the same as in the standard LSTM that traverses from time $t = 1$ to T. However, in the case of the backward layer, the input sequence is fed from t = T to 1. Therefore, a mathematical representation of the LSTM cell in the forward layer will remain the same whereas mathematical expressions of the LSTM cell in the backward layer denoted by the leftward arrow ($\leftarrow$) at time t can be written as:

$$\vec{F} = \partial\left(W^{\overleftarrow{f}} X_t + W^{\overleftarrow{f}} R_{t+i}\right) \tag{7}$$

$$\vec{I} = \partial(W^{\overleftarrow{i}} X_t + W^{\overleftarrow{i}} R_{t+i}) \tag{8}$$

$$\vec{O} = \partial W^{\overleftarrow{o}} X_t + W^{\overleftarrow{o}} R_{t+i} \tag{9}$$

$$\vec{G} = tanh(W^{\overleftarrow{g}} X_t + W^{\overleftarrow{g}} R_{t+i}) \tag{10}$$

$$\vec{C_t} = \vec{F} \odot \vec{C_{t-1}} + \vec{I} \odot \vec{G} \tag{11}$$

$$\vec{R_t} = \vec{O} \odot tanh(\vec{C_t}). \tag{12}$$

At each time step of bidirectional LSTM, two outputs will be generated, one by forwarding the LSTM layer ($R_t$) and one by the backward LSTM layer ($\vec{R_t}$). These two outputs will be

merged to get the output $y_t$. Final output Y will be expressed as the combination of the outputs generated by individual LSTM layers.

$$Y_t = R_t + \vec{R}_t \tag{13}$$

$$Y = y_{t-1} + y_t + y_{t+1} \tag{14}$$

Figure S9 shows the working of three-time step bidirectional LSTM. Both forward and backward layers work independently in opposite directions to preserve contextual information. The output will then be processed, based on both past and future values.

### Construction of stacked LSTM model

The Stacked LSTM has four layers: an input layer, a first LSTM layer, a second LSTM layer, and an output layer with 32, 16, 16, and 1 neuron respectively. The input vector to the model is X that consists of features $(u_t..u_{t-1}, v_t..v_{t-1}, ws_t..ws_{t-1}, wd_t..wd_{t-1}, p_{t-n}..p_{t-1})$ of feature set S(t) with past n observations whereas feature($p_t$), the wind power, is our output vector Y. The first LSTM layer uses the Rectified Linear Unit (Relu) as an activation function and returns the processed sequence to the second LSTM layer. Finally, the predicted sequence is passed to the densely connected output layer. The weight initializer parameter of the model is set as 'RandomUniform' which initiates the network weights with random and uniformly distributed values in the range of 0 to 0.9. Root mean square propagation (RMSprop) is used as a model optimizer with a learning rate of 0.001. The model is trained with epochs = 5 and a batch size of 32. Figure 1 describes the architecture and flow of the stacked LSTM model.

### Construction of bidirectional LSTM model

The bidirectional Stacked LSTM used in this study has four layers: the input layer, the first bidirectional LSTM layer, the second bidirectional LSTM layer, and an output layer with 64, 32, and 1 neuron respectively. The remaining tuning parameters like weight initializer, activation function, optimizer algorithm, learning rate, batch size, and the number of epochs are the same as the stacked LSTM model. The architecture of the proposed bidirectional LSTM can be seen in Fig. 2. In this model, simple LSTM layers are replaced with bidirectional LSTM layers.

### Forecast accuracy

To perform more realistic forecasts, a step time-series cross-validation technique as indicated in the research (*Zhang et al., 2022*) is used. Preserving the time dependency aspect of the time series data, on each iteration, this method uses the previously observed values as a training set and the next unseen observation as a test set. To frame data for one-step cross-validation, a window of previous n observations was included as a feature at the data preprocessing stage. So besides splitting the dataset into training and test data, a one-step time-series cross-validation is also performed to enhance the forecast accuracy.

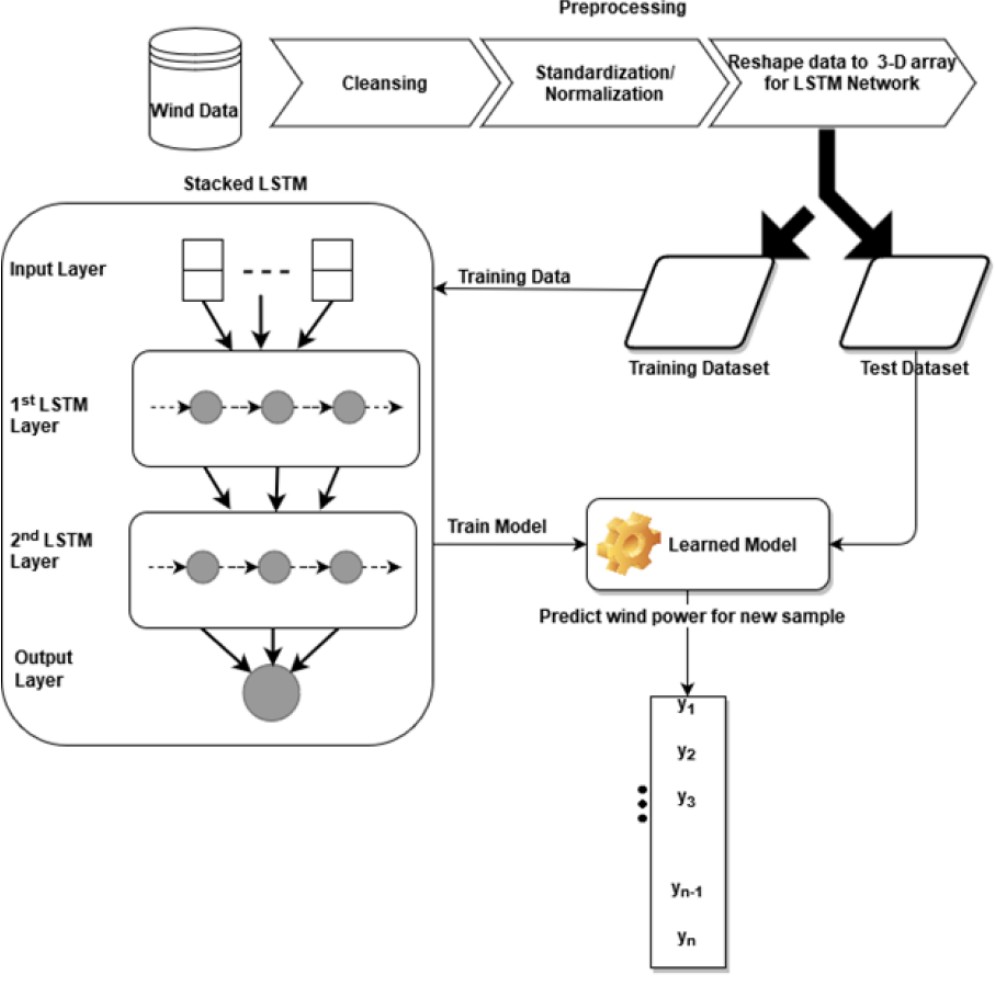

**Figure 1**  **Architecture of stacked LSTM.**

## Performance parameters

Metrics used to evaluate the performance of the developed LSTM models are mean absolute error (MAE), root mean square error (RMSE), and standard deviation error (SDE) defined as

$$MAE = \left| \frac{\sum (Xactual.i - Xmodel.i)}{n} \right| \tag{15}$$

$$RMSE = \sqrt{\frac{\sum (Xactual.i - Xmodel.i)^2}{n}} \tag{16}$$

$$\left( SDE = \sqrt{\frac{1}{M} \sum_{i=1}^{M} (Error - MeanError)^2} \right) \tag{17}$$

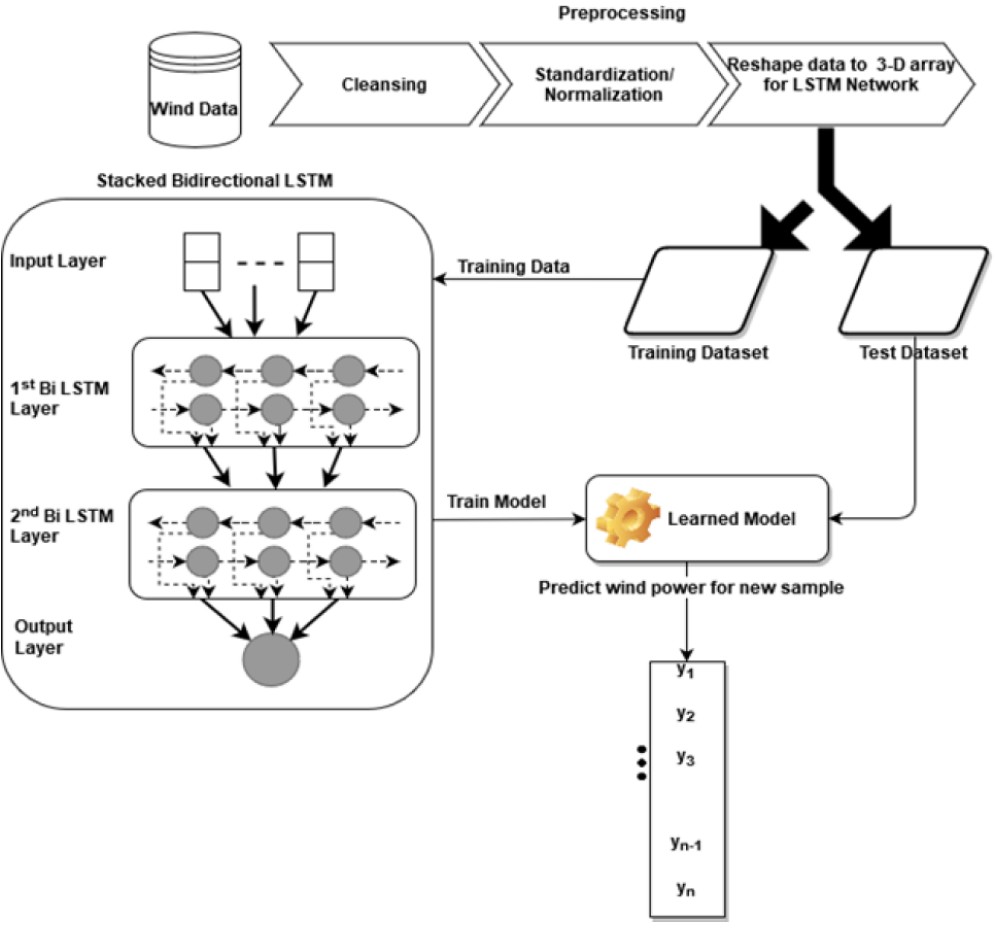

**Figure 2 Architecture of bidirectional LSTM.**

where $X_{actual}$ is a vector consisting of actual values of wind power and $X_{model, i}$ is the model predicted value at time/place *i*. The objective of the developed model is to minimize the values of these parameters to attain higher performance. Whereas $Wp_{actual}$ is the desired value of power while $Wp_{predicted}$ is the predicted power value provided by the proposed technique. $Error = Wp_{actual} - Wp_{predicted}$, while the average value of error is called mean error.

## RESULTS AND DISCUSSION

To quantitate performance evaluation of the developed LSTM model, three different metrics such as MAE, RMSE, and SDE have been utilized. Five wind farm data is used as a dataset to compare the results with the existing techniques of *Zameer et al. (2017)* and *Qureshi et al. (2017)*.

### Stacked LSTM model

The model was implemented using 80% of the data as training and the remaining 20% for testing purposes. Table 1 shows the parameter settings applied to train the model.

**Table 1 Stacked LSTM hyper parameter settings.**

| S No. | Parameter | Setting |
|---|---|---|
| 1 | Number of Epochs | 5 |
| 2 | Batch Size | 32 |
| 3 | n (No of past values to be included) | 5 |
| 4 | Loss function | Logcosh |
| 5 | Optimizer | Rmsprop |
| 6 | Learning rate | 0.001 |

**Notes.**
Hyperparameters for both stacked and bidirectional LSTM are the same.

**Table 2 Results of stacked LSTM on training (TR) and testing (TS) data.**

| Dataset | RMSE (TR) | RMSE (TS) | MAE (TR) | MAE (TS) | SDE (TR) | SDE (TS) |
|---|---|---|---|---|---|---|
| Wind Farm 1 | 0.0935 | 0.0856 | 0.0614 | 0.0530 | 0.2444 | 0.1553 |
| Wind Farm 2 | 0.1042 | 0.0717 | 0.0614 | 0.0464 | 0.2709 | 0.1797 |
| Wind Farm 3 | 0.1197 | 0.0578 | 0. 0640 | 0.0403 | 0.2965 | 0.1050 |
| Wind Farm 4 | 0.1095 | 0.1063 | 0.0787 | 0.0712 | 0.0288 | 0.1475 |
| Wind Farm 5 | 0.1193 | 0.1022 | 0.0732 | 0.0589 | 0.2995 | 0.1998 |

From the performance perspective, it has been observed that the proposed techniques achieved optimum results with only five epochs, whereas existing techniques used 150 to 500 epochs. Training and testing results obtained from the model are shown in Table 2. During the training phase, optimum results were observed when the past five observations were included in the feature set. So, the value of *n i.e.,* no of past values to be included in the feature set is tuned as five.

Figures 3 through 7 shows the results of actual *vs* predicted hourly wind power for all the wind farms. The results show that there is a close match between the actual and predicted wind power. Both actual and predicted wind power curves almost follow the same pattern. It has been observed through results that LSTM is a robust technique for time sequence problems. Results also support the fact that deep neural networks work best when the dataset is large as in our case dataset for Wind Farm 3 is comparatively large, therefore, significant improvement in terms of error reduction is observed.

## Bidirectonal LSTM model

As for Stacked LSTM, the dataset for the bidirectional LSTM model is also divided into a proportion of 80% and 20% for training and testing respectively. The same parameter settings of stacked LSTM are used for bidirectional LSTM. Table 3 shows both training and testing results when bidirectional LSTM is applied to the dataset.

Figures 8 through 12 show the results of actual *vs* predicted wind power on each wind farm dataset when bidirectional LSTM is used. Results demonstrate that bidirectional LSTM performs slightly better than stacked LSTM, especially in the case of Wind Farm 3, the performance difference increases concerning the proportion of data available for training. So, results can be improved when applied to large datasets.

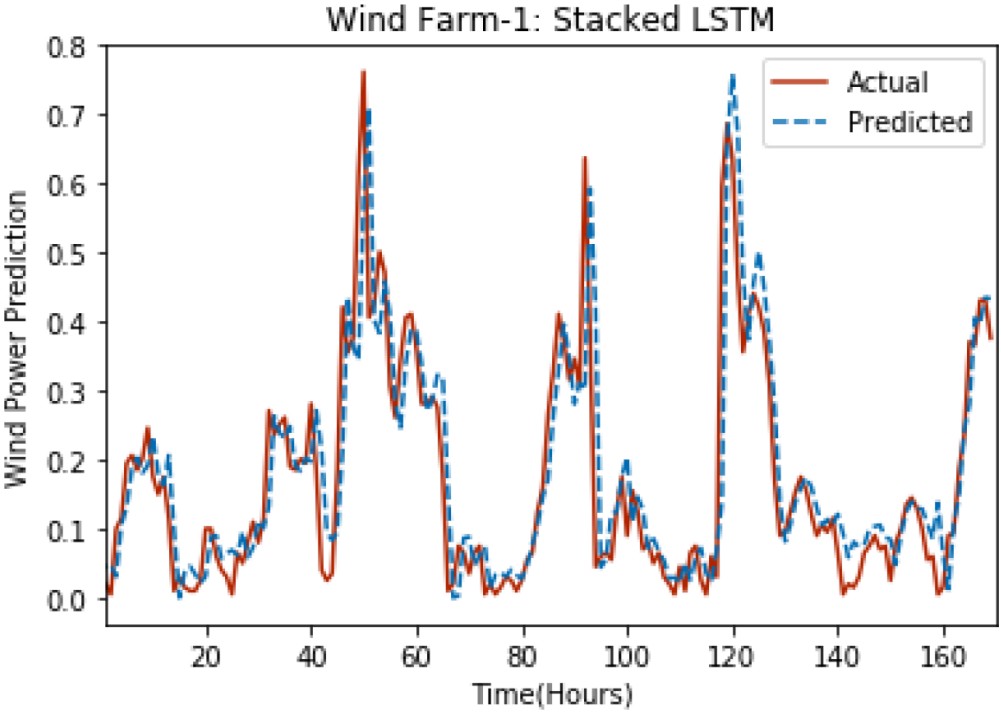

**Figure 3** **Plot of stacked LSTM for Wind Farm 1.**

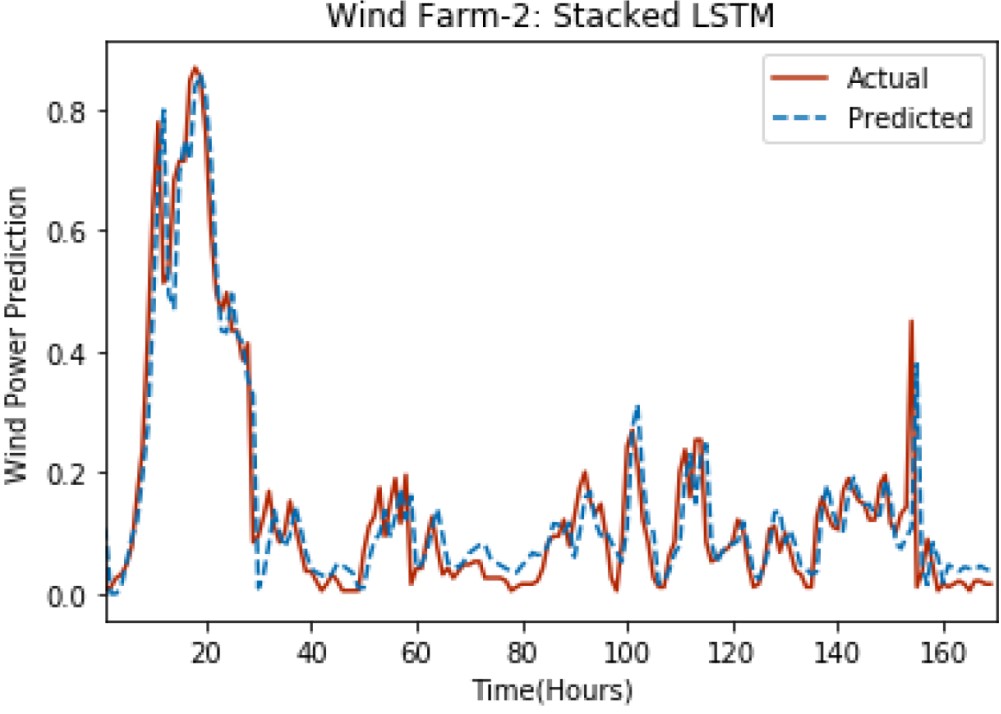

**Figure 4** **Plot of stacked LSTM for Wind Farm 2.**

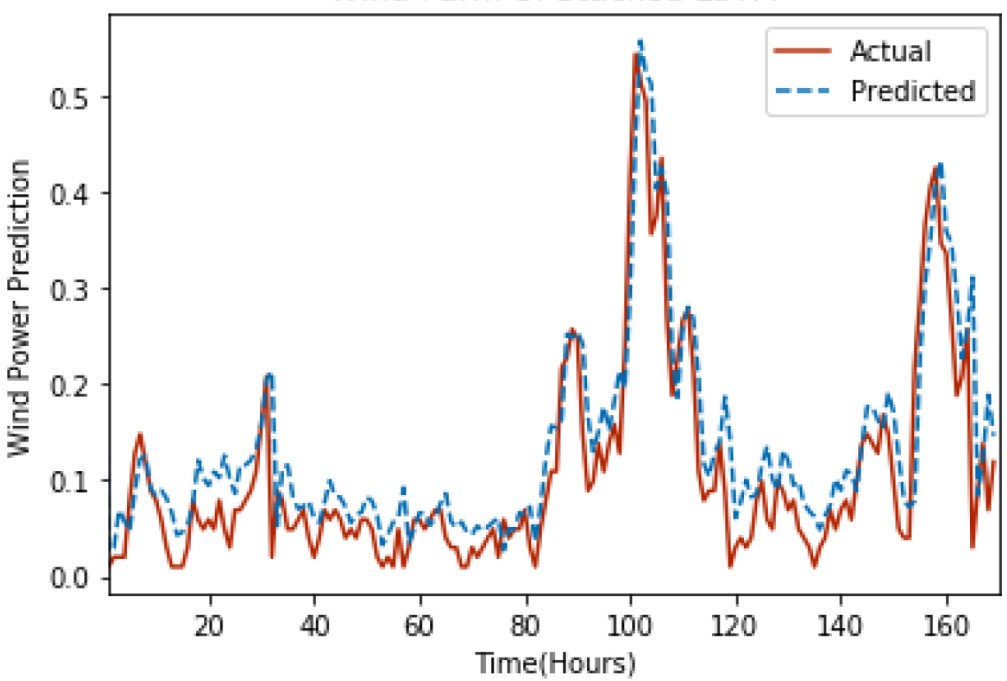

**Figure 5  Plot of stacked LSTM for Wind Farm 3.**

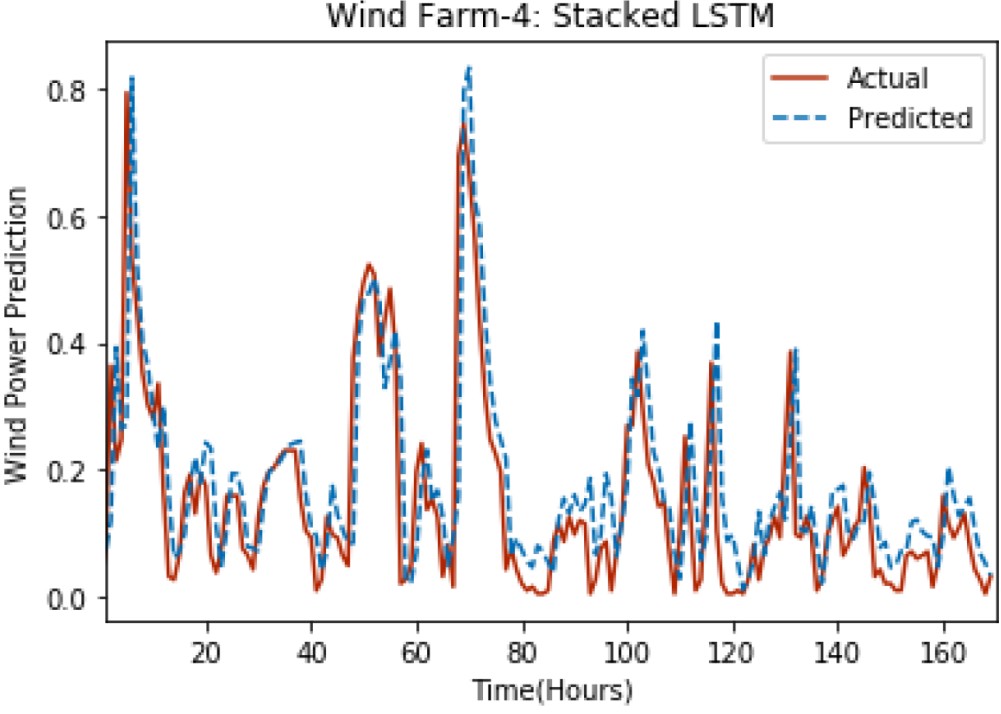

**Figure 6  Plot of stacked LSTM for Wind Farm 4.**

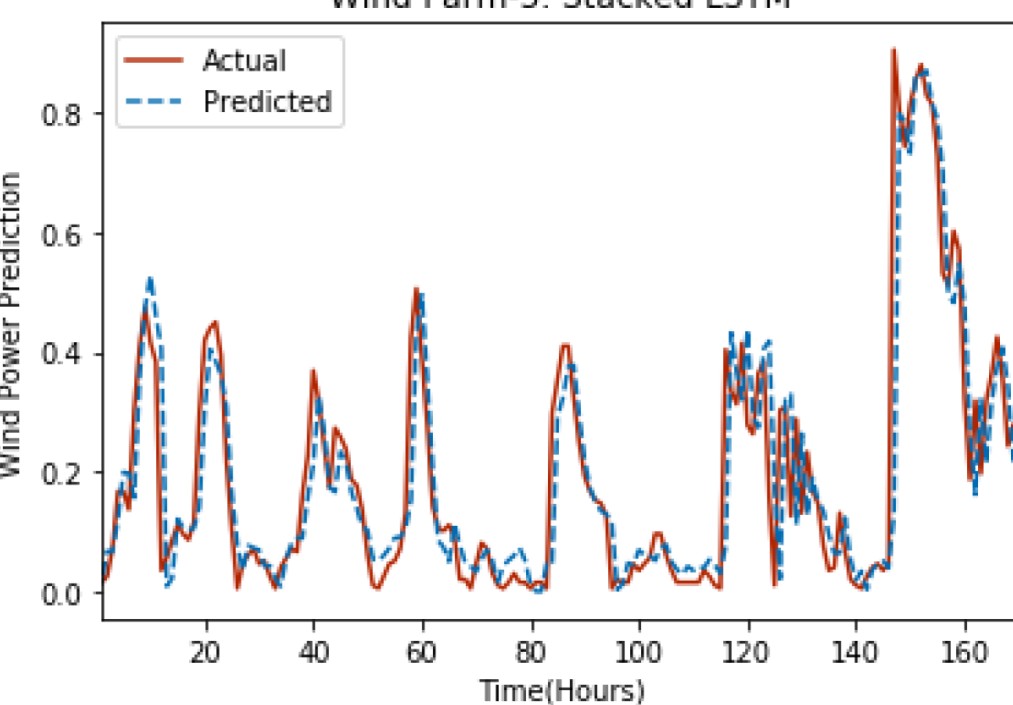

**Figure 7**  Plot of stacked LSTM for Wind Farm 5.

**Table 3**  Results of bidirectional LSTM on training (TR) and testing (TS) data.

| Dataset | RMSE (TR) | RMSE (TS) | MAE (TR) | MAE (TS) | SDE (TR) | SDE (TS) |
|---|---|---|---|---|---|---|
| Wind Farm 1 | 0.0933 | 0.0836 | 0.0606 | 0.0505 | 0.2406 | 0.1550 |
| Wind Farm 2 | 0.1029 | 0.0747 | 0.0632 | 0.0473 | 0.2787 | 0.1869 |
| Wind Farm 3 | 0.1196 | 0.0539 | 0.0790 | 0.0385 | 0.2854 | 0.0962 |
| Wind Farm 4 | 0.1120 | 0.1016 | 0.0774 | 0.0643 | 0.2685 | 0.1403 |
| Wind Farm 5 | 0.1179 | 0.0999 | 0.0730 | 0.0587 | 0.2951 | 0.1996 |

## Results comparison with existing techniques

In this section, the results of the proposed models are compared with the existing techniques of GPeANN (*Zameer et al., 2017*) and DNN-MRT (*Qureshi et al., 2017*). Table 4 shows the results in terms of RMSE, MAE, and SDE of existing and proposed models on training and test datasets. Figures 13 to 15 shows a comparison of the results of different error metrics measured against each wind farm. The comparison clearly shows that the Stacked LSTM and bidirectional LSTM outperformed the other two methods in terms of RMSE and MAE. However, in the case of SDE, the proposed models only outperformed Windfarm 3. The reason could be that in Wind Farm 3, the variation of wind power is generally lower than in other datasets. Furthermore, the author calculates SDE as the standard deviation

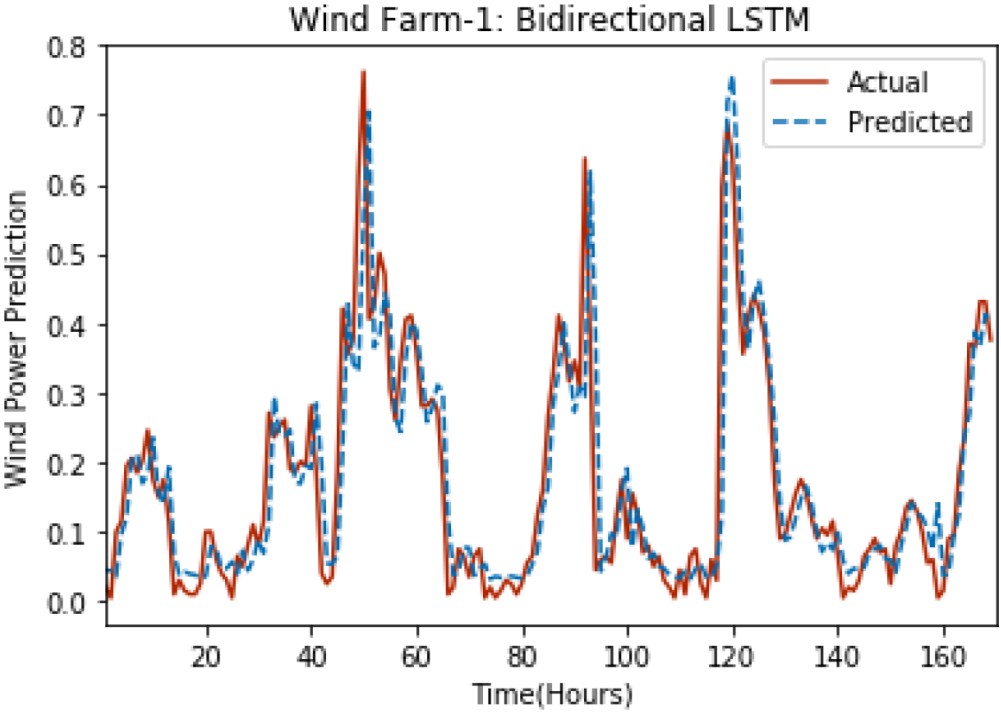

**Figure 8** Plot of bidirectional LSTM for Wind Farm 1.

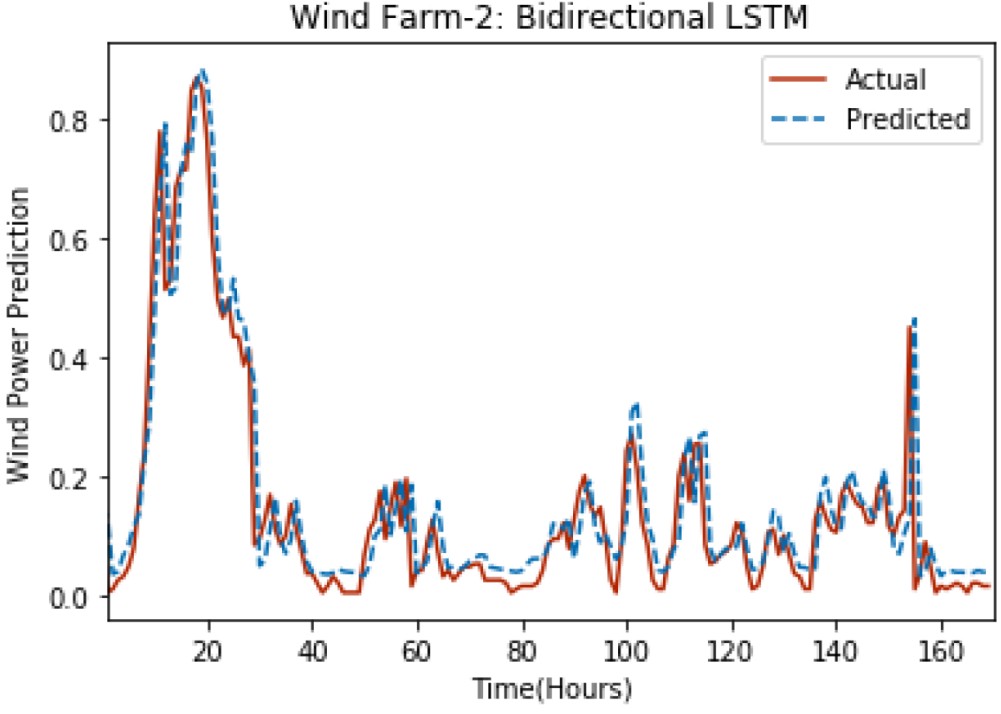

**Figure 9** Plot of bidirectional LSTM for Wind Farm 2.

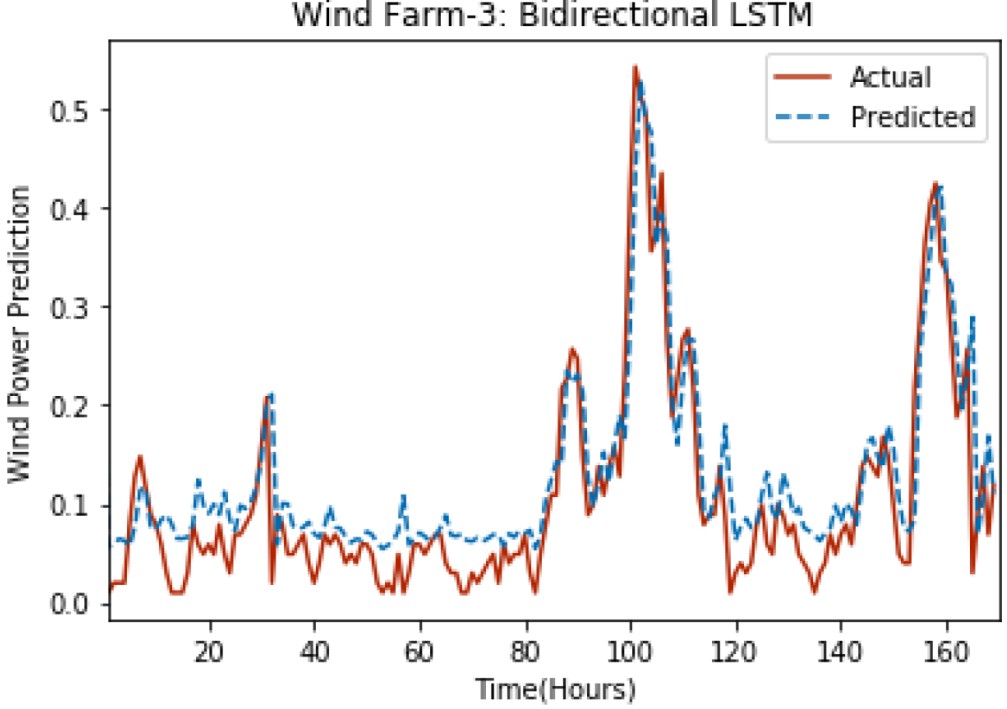

**Figure 10** Plot of bidirectional LSTM for Wind Farm 3.

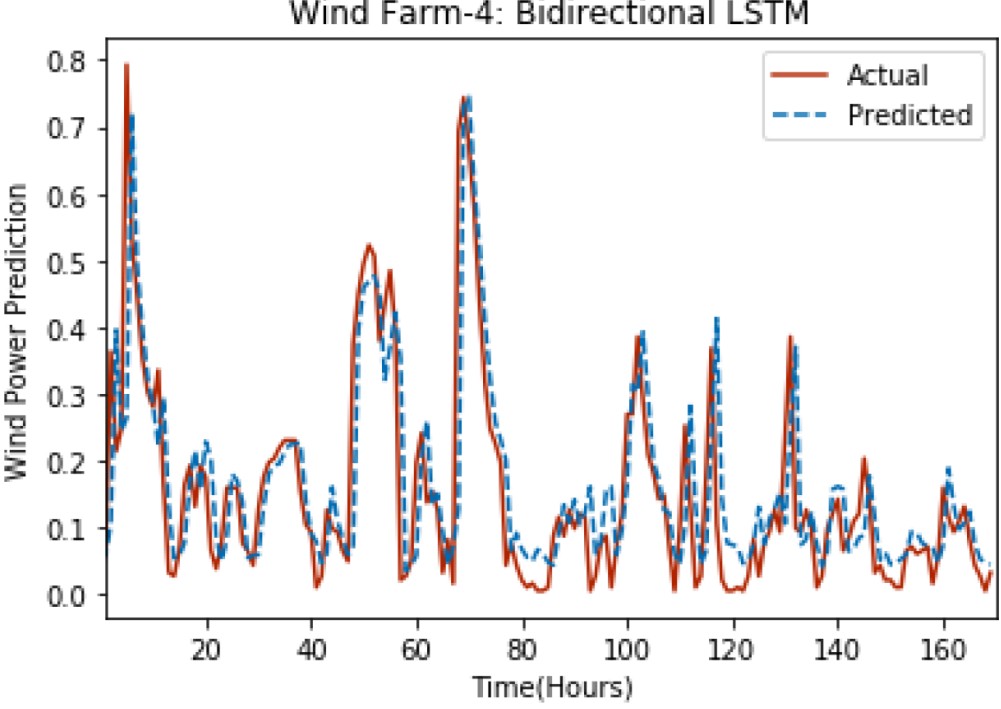

**Figure 11** Plot of bidirectional LSTM for Wind Farm 4.

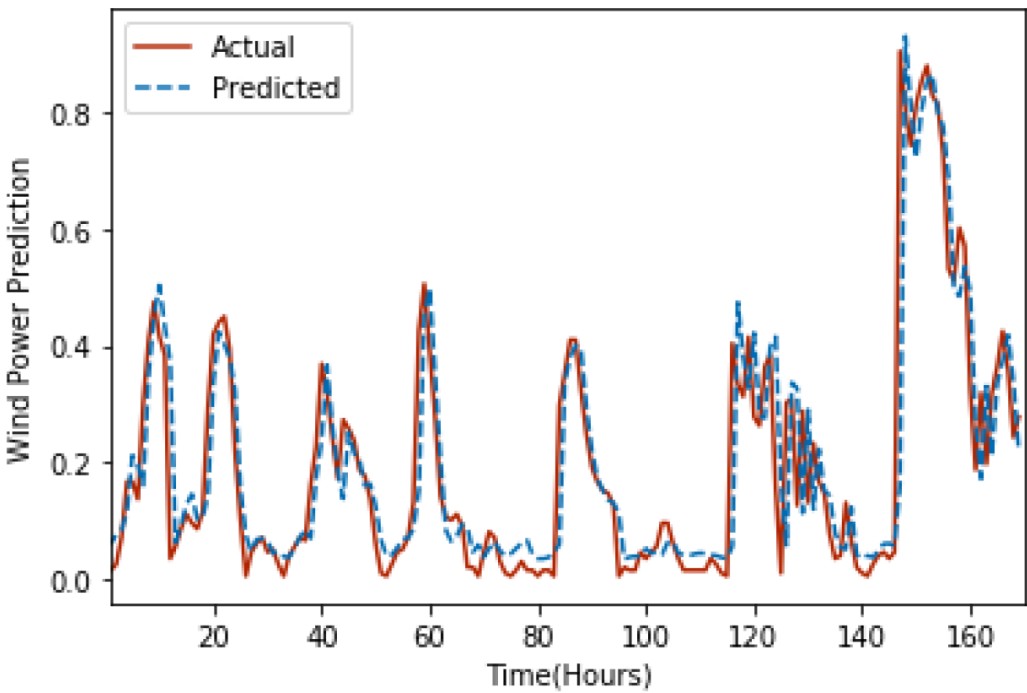

**Figure 12** Plot of bidirectional LSTM for Wind Farm 5.

**Table 4** Comparison of results with existing techniques.

|  | Error measures | Training data | | | | Test data | | | |
|---|---|---|---|---|---|---|---|---|---|
|  |  | GPeANN | DNN-MRT | Stacked LSTM | Bidirectional LSTM | GPeANN | DNN-MRT | Stacked LSTM | Bidirectional LSTM |
| Wind Farm 1 | RMSE | 0.0869 | 0.0721 | 0.0935 | 0.0933 | 0.0966 | 0.0939 | 0.0856 | 0.0836 |
|  | MAE | 0.0575 | 0.0508 | 0.0614 | 0.0606 | 0.0643 | 0.0658 | 0.0530 | 0.0505 |
|  | SDE | 0.0868 | 0.0719 | 0.2444 | 0.2406 | 0.095 | 0.0929 | 0.1553 | 0.1550 |
| Wind Farm 2 | RMSE | 0.0975 | 0.0809 | 0.1042 | 0.1029 | 0.1157 | 0.1032 | 0.0717 | 0.0747 |
|  | MAE | 0.0623 | 0.0564 | 0.0640 | 0.0632 | 0.0739 | 0.0713 | 0.0464 | 0.0473 |
|  | SDE | 0.0957 | 0.0782 | 0.2709 | 0.2787 | 0.1152 | 0.1025 | 0.1797 | 0.1869 |
| Wind Farm 3 | RMSE | 0.1071 | 0.0903 | 0.1197 | 0.1196 | 0.135 | 0.1207 | 0.0578 | 0.0539 |
|  | MAE | 0.0694 | 0.0607 | 0.0787 | 0.0790 | 0.0874 | 0.0825 | 0.0403 | 0.0385 |
|  | SDE | 0.1071 | 0.0897 | 0.2965 | 0.2854 | 0.1329 | 0.1203 | 0.1050 | 0.0962 |
| Wind Farm 4 | RMSE | 0.1061 | 0.0838 | 0.1095 | 0.1120 | 0.1118 | 0.1036 | 0.1063 | 0.1016 |
|  | MAE | 0.0681 | 0.0603 | 0.0732 | 0.0774 | 0.078 | 0.0748 | 0.0712 | 0.0643 |
|  | SDE | 0.1061 | 0.0827 | 0.0288 | 0.2685 | 0.1118 | 0.1036 | 0.1475 | 0.1403 |
| Wind Farm 5 | RMSE | 0.1135 | 0.1008 | 0.1193 | 0.1179 | 0.1203 | 0.1156 | 0.1022 | 0.0999 |
|  | MAE | 0.0715 | 0.0705 | 0.0731 | 0.0730 | 0.077 | 0.0804 | 0.0589 | 0.0587 |
|  | SDE | 0.1135 | 0.0952 | 0.2995 | 0.2951 | 0.119 | 0.1137 | 0.1998 | 0.1996 |

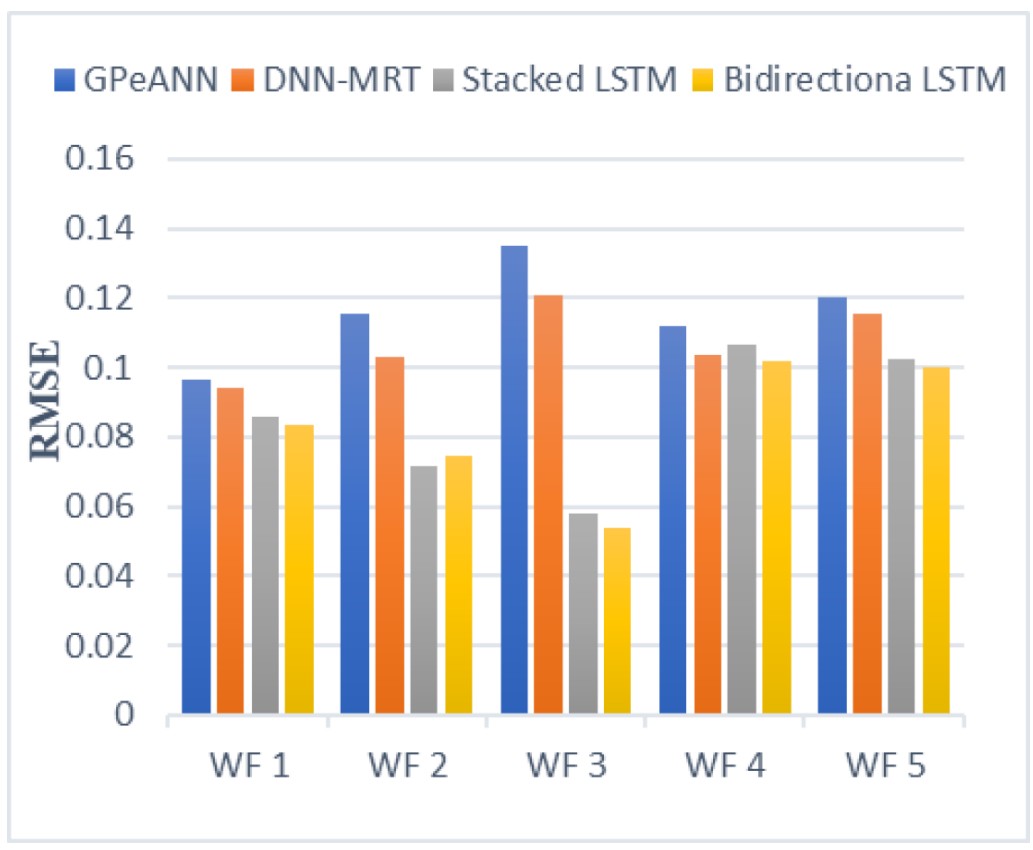

**Figure 13** **Root mean squared error of all models.**

of predicted wind power values, which is highly influenced by the variation of true wind power.

It is also noteworthy that the proposed model effectively generalizes patterns from the training set to the test set. This indicates that the model has successfully learned the underlying patterns and can apply them to new, unseen data. The utilization of regularization techniques during training of the proposed models prevents overfitting, thereby enhancing the model's ability to generalize to unseen data. Furthermore, the LSTM's inherent architecture and ability to capture long-term dependencies contribute to better performance on the test set. The model has learned intricate temporal relationships within the data that aid in making more accurate predictions. Moreover, the test set may have characteristics or patterns that closely align with the training set, allowing the proposed LSTM model to perform well on unseen data that share similarities with the training data.

## CONCLUSIONS, LIMITATIONS AND FUTURE WORK

The present work proposed the use of the LSTM technique to predict wind power. Data preprocessing was performed using statistical methods like simple arithmetic means and normalization. To enhance the performance of the model, the previous five power values were also utilized as well. The simple and bidirectional stacked LSTM models were
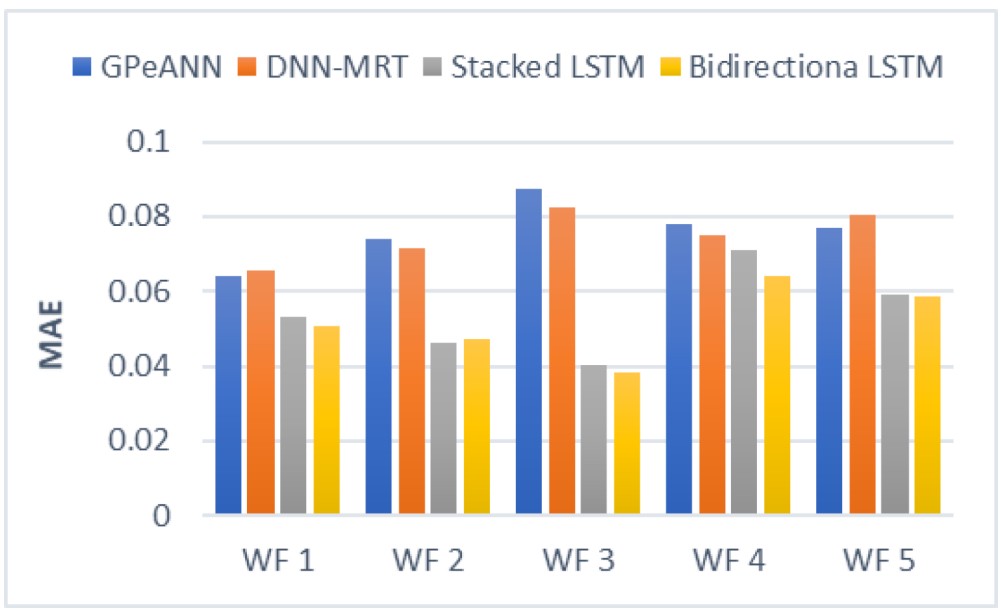

Figure 14 Mean absolute error of all models.

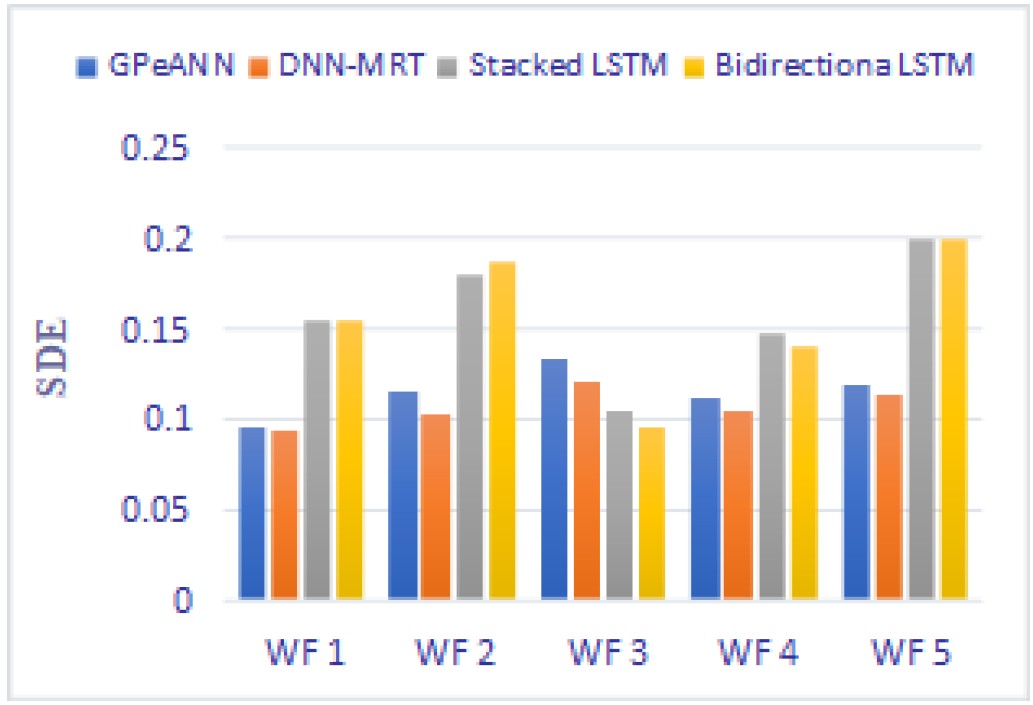

Figure 15 Standard deviation of error of all models.

implemented and tested on five wind farms data obtained from the Kaggle website. The proposed technique used the DNN-based features of LSTM to exploit the wind power and related metrological multivariate time series data. The results were compared with the existing state-of-the-art techniques, where metrics *e.g.*, RMSE, MAE, and SDE showed that the proposed models give a comparatively good performance. In the best case, it reduces the RMSE and MAE values to almost fifty percent. Results show that the accuracy of the proposed models increases significantly when applied to larger datasets. Existing approaches used complex architectures with a greater number of neurons, and neural network layers, and required more processing power to achieve the results, whereas the proposed LSTM model performs better with just two LSTM layers and requires minimum processing power to achieve comparable results. From the results, it can be concluded that the proposed models outperformed the existing models both in terms of performance and accuracy.

Accurately predicting wind speed and power generation for the next hour offers numerous benefits for the wind industry, impacting various aspects including such as: improving grid integration and stability, in optimized energy trading and market participation, in operational and maintenance optimization, in overall cost reduction and various other environmental benefits. For a detailed description, please see related works such as *Yang et al. (2023)*, *Rama, Hur & Yang (2024)*, *Saini et al. (2023)* and, *Ying et al. (2023)*.

The dataset used in the implementation of models is specific to different wind farms in Europe. However, proposed LSTM models can be generalized to be used for other wind farm data. This study did not conduct a robust statistical analysis to validate the superiority of the suggested methodology over current approaches. Therefore, future work can plan to conduct a thorough statistical examination to ensure the reliability and significance of the performance differences. Additionally, the application of the approach to novel wind farm datasets and consideration of potential obstacles in diverse settings is an important endeavor lacking in this research. Future research could explicitly address these concerns by conducting comprehensive experiments on various wind farm datasets and identifying and mitigating challenges that may arise in different settings.

Besides, there are multiple limitations in the use of the matrices such as MAE, SDE, and RMSE. MAE treats all errors equally without considering their direction. It does not penalize large errors more than small ones, which may not be appropriate in situations where certain errors have more significant consequences. Furthermore, MAE is sensitive to outliers. A single large error can disproportionately influence the overall performance metric. Being an absolute measure, MAE does not provide information about the direction of the errors (overestimation or underestimation). The metric SDE is sensitive to the scale of the data. It may not be directly comparable across datasets with different units or ranges. Like MAE, SDE does not distinguish between overestimation and underestimation, providing a measure of overall error variability but lacking directional information. Finally, RMSE gives higher weight to large errors due to the square term. This makes it more sensitive to outliers compared to MAE. Like MAE, RMSE does not provide information about the direction of errors, making it challenging to interpret the nature of the model's mistakes.

Another possibility of future work is the integration of attention mechanisms within our LSTM models. Attention mechanisms allow the model to assign different weights to different parts of the input sequence, providing a more transparent view of which elements contribute more significantly to the final prediction. By visualizing these attention weights, insights into the aspects of the input data that influence the model's decisions can be viewed. Additionally, exploring techniques that ascertain the value of features during the model's decision-making could be taken into consideration. This involves conducting feature importance analysis to identify which input features have the most impact on the model's predictions. This could make an LSTM model more interpretable and informative.

Besides prediction for the following hour, the work can be further extended to facilitate medium to long-range wind power forecasting with adjustments to the model parameters. To further improve the accuracy, larger datasets are required. In the future, this work can be further extended to facilitate medium to long-range wind power forecasting with adjustments to the model parameters and devising hybrid approaches by combining different LSTM variants with other state-of-the-art techniques.

This study explored four different methods for wind power forecasting, comparing their results against each other and the baseline article. While evaluating the MAE, the wind power data was normalized between 0 and 1 for consistency. However, this normalization means the reported MAE values, ranging from 4% to 8%, might not directly translate to percentage changes in actual power values. Therefore, a key question arises: is this error range adequate for hourly wind power forecasting? Answering this definitively necessitates further research that compares our findings with other studies using similar data and normalization techniques. Such a comparison would offer a clearer understanding of our model's performance in real-world applications.

Another future work possibility could be a comparison to encompass not only current cutting-edge methods but also alternative neural network architectures like Gated Recurrent Units (GRUs). Integrating a comparative analysis between LSTM and GRUs would enrich forthcoming research, offering deeper insights into the distinct advantages attributed to LSTM and its comparative edge over alternative architectures. This expanded analysis will be a valuable direction for future investigations in the realm of wind power prediction, fostering a more comprehensive understanding of neural network performance in this domain.

### Funding

This work is supported by the EIAS (Emerging Intelligent Autonomous Systems) Data Science Lab, Prince Sultan University, KSA. Prince Sultan University supported the Article Processing Charges for this article. The funders had no role in study design, data collection and analysis, decision to publish, or preparation of the manuscript.

### Grant Disclosures

The following grant information was disclosed by the authors:

EIAS (Emerging Intelligent Autonomous Systems) Data Science Lab, Prince Sultan University, KSA.

## Competing Interests

The authors declare there are no competing interests.

## Author Contributions

- Mehmood Ali Khan conceived and designed the experiments, performed the experiments, performed the computation work, prepared figures and/or tables, and approved the final draft.
- Iftikhar Ahmed Khan conceived and designed the experiments, analyzed the data, performed the computation work, prepared figures and/or tables, and approved the final draft.
- Sajid Shah performed the experiments, authored or reviewed drafts of the article, and approved the final draft.
- Mohammed EL-Affendi analyzed the data, authored or reviewed drafts of the article, and approved the final draft.
- Waqas Jadoon conceived and designed the experiments, performed the experiments, analyzed the data, authored or reviewed drafts of the article, and approved the final draft.

## Data Availability

The data is available at Kaggle: https://www.kaggle.com/competitions/GEF2012-wind-forecasting/overview.

The code is available at Zenodo: Khan, M. A. (2024). LSTM Prediction Code and Dataset extracated from Kaggle dataset. Zenodo. https://doi.org/10.5281/zenodo.10477563.

## Supplemental Information

Supplemental information for this article can be found online at http://dx.doi.org/10.7717/peerj-cs.1949#supplemental-information.

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
