# Peer review of "Short-term wind power forecasting through stacked and bi directional LSTM techniques"

_PeerJ Computer Science, doi:10.7717/peerj-cs.1949_

## Round 0.1 · original submission · Major Revisions

Referee 4 requested citations to 6 articles. Only 2 of these 6 articles are about wind. Therefore, I recommend citing only these two articles, if any.

Additional comparisons should be made with new deep-learning algorithms to confirm the performance of the proposed technique. In this form, the discussion part seems weak.

**Language Note:** PeerJ staff have identified that the English language needs to be improved. When you prepare your next revision, please either (i) have a colleague who is proficient in English and familiar with the subject matter review your manuscript, or (ii) contact a professional editing service to review your manuscript. PeerJ can provide language editing services - you can contact us at copyediting@peerj.com for pricing (be sure to provide your manuscript number and title). – PeerJ Staff

Reviewer 1 ·

Basic reporting

All specialists in their respective disciplines were able to read and comprehend it because it was written in formal, impartial, and objective English. This was done to ensure that the material could be read and understood by all professionals. This document contains sufficient background information as well as relevant references. The introduction, summary, and conclusion all present the information coherently and accurately. In addition to proper labelling, it also had a header, so it met all requirements. It includes both the methods and outcomes. The results are entirely reproducible, and the code is presented in a format that makes it simple to execute. It is capable of meeting the requirements for acceptance into the technically suitable area.

Experimental design

You may feel comfortable submitting it because it is an original piece that suits the "airms and scope" of the journal for which you are writing. The problem with the investigation is blatantly obvious, so the solution must be both timely and significant. In addition, the response fills a void in the existing body of knowledge; comments on this aspect of the work's contribution to the body of knowledge should be provided. This fills a previously existing lacuna in the relevant literature. It complies not only with the "airms and scope" of the publication, but also with the ethical standards mentioned earlier in this sentence. Detailed explanations and simulated plans are provided for the steps that must be followed to complete an assignment. The training and formula explanations for the stacked bidirectional model are very appropriate.

Validity of the findings

The findings are conveyed in a comprehensive and detailed manner that is also original and innovative. The discussion is restricted to the study findings that support the aforementioned topics, and they are related to the primary research queries. Given that both the dataset and the code are publicly accessible and made available for use, it is straightforward to reproduce the study's findings. It has been the subject of a review employing statistical techniques and methodologies. The benefits of incorporating this new corpus of literature into the existing canon have been enumerated in an approachable and concise manner.

Additional comments

Formulas and tables are included in the text; nevertheless, it would have been helpful if readers were also provided with some visually appealing graphics.

·

Basic reporting

As per the study Researchers have been actively exploring renewable energy sources as a solution to global warming, fossil fuel depletion, and increasing electricity demand. Among these sources, wind power has gained popularity due to its eco-friendly nature, cleanliness, viability, and cost-effectiveness. However, wind power generation relies on the variable and unpredictable nature of wind speed, necessitating accurate forecasting models for the industry. Several machine learning (ML) techniques have been developed to predict wind power and speed using weather data and historical records.
In the introduction, the authors provide a clear rationale for the research, emphasizing the importance of accurate wind power prediction for effective energy management. They outline the limitations of traditional forecasting methods and introduce LSTM models as a promising solution due to their ability to capture long-term dependencies and handle temporal data effectively. The introduction demonstrates a strong understanding of the subject matter and effectively engages the reader.

Experimental design

The results and discussion section presents the findings obtained from the application of the stacked and bi-directional LSTM models. The authors present numerical evaluations, performance metrics, and comparative analyses with other forecasting methods to validate the effectiveness of their proposed approach. The results are well-organized, visually represented through graphs and tables, and interpreted standard deviation of error, Standard Deviation of Error of all Models, in a clear and meaningful manner.

Validity of the findings

Author has well-articulated the research study with relevant references and research and that data is well utilized in defining the final recommendations.
Author has touched how the ancient city structure evolved into IoT smart city. The metrics published are up to the mark and clear. The dataset used in the implementation of models is specific to different wind farms in Europe. 20 However, proposed LSTM models can be generalized to be used for other wind farm data
This research has been supported by EIAS Data Science Lab, Prince Sultan University, KSA. The 30 authors would like to thank EIAS Data Science Lab and Prince Sultan University
The size of the analysis number is high and accurate. Community engagement model, targeted citi count and size. Who are contributing to the smart cities. Over years how many publications are existing on smart city concept.
Strength: Vast research and well described the proposed solution. In the future, this work can be further extended 24 to facilitate medium to long-range wind power forecasting with adjustments to the model 25 parameters and devising hybrid approaches by combining different LSTM variants with other 26 state-of-the-art techniques.

Reviewer 3 ·

Basic reporting

The paper demonstrates clear and professional English writing. It is well-organized with appropriate inclusion of figures, tables, and raw data. The literature references sufficiently provide background and context for the research. The study is both original and meaningful. The methods are adequately described, facilitating replication. However, there are a few typos present, and some subsections have incorrect numbering, which should be addressed and corrected.

Experimental design

1. Section 2.1, line 41-42: the dataset provided by the Kaggle competition consists of seven wind farms in total. However, this paper only introduces and utilizes five of them. The author should clarify the reason for selecting specific data subsets.
2. Section 2.3, line 39-42: The figure and its description are somewhat misleading. The unfolded RNN's output should encompass a longer sequence, rather than solely R_t-1, R_t, and R_t+1. Additionally, the figure reference number provided here is incorrect.
3. In section 2.9, the standard deviation error (SDE) is defined in formula 17. The author should elaborate on the rationale behind this definition, as it represents the deviation of the predicted value from the actual mean value. Alternatively, if the intent is to measure the standard deviation of the error, formula 17 should be corrected.
4. In section 2.10, line 14-16, the claim that using a much simpler model would yield equal or superior performance compared to complex models lacks sufficient supporting explanation. Further elaboration is necessary.
5. Table II - VI: The author should discuss why the proposed models generally exhibit better performance on the test set compared to the training set, whereas the baseline models perform better on the training set. An exploration of these discrepancies is needed.
6. Section 2.12, line 20-22: The observed better performance of SDE on Wind Farm 3 is not due to the dataset size. Wind Farm 3 is not significantly larger than the other datasets. The primary reason lies in the fact that in Wind Farm 3, the variation of wind power is generally lower than in other datasets. Furthermore, the author calculates SDE as the standard deviation of predicted wind power values, which is highly influenced by the variation of true wind power.

Validity of the findings

The research question addressed in this paper is significant and relevant to the field. The study presents findings that contribute to the existing knowledge in the domain. However, there are several aspects that need further consideration and clarification. The rationale behind the chosen dataset subset should be explained, and the definition of the standard deviation error (SDE) requires further explanation. The claim regarding simpler models' performance needs more evidence and explanation. The performance comparison between proposed and baseline models and the attribution of better SDE performance require objective analysis. Addressing these points will enhance the validity of the findings.

Additional comments

The paper presents valuable findings that demonstrate proficiency in research design, methodology, and analysis. The writing style is clear and professional. However, there are a couple of points that require attention:
1. In the abstract (line 24-26), the author should refrain from claiming to have proposed the LSTM architecture and instead state that they proposed a method based on LSTM.
2. In Section IV, line 4-5, the term "data preprocessing" should be used instead of "feature selection" since normalization is not considered a feature selection technique.

Reviewer 4 ·

Basic reporting

Please see the attached file.

Experimental design

Please see the attached file.

Validity of the findings

Please see the attached file.

Additional comments

Please see the attached file.

Annotated reviews are not available for download in order to protect the identity of reviewers who chose to remain anonymous.

Reviewer 5 ·

Basic reporting

The authors have presented the short term wind power forecasting using deep learning techniques. The presented techniques like LSTM are not new. Many authors have done a lot of work by using these techniques. I have few comments which are mentioned below:
1. The authors need to suggest some new technique in their paper for example ensemble of the suggested technique to show some innovation. Authors need to work on this and include or discuss some new method which is unique and has not been implemented yet.
2. The authors need to include some more latest references from year 2023 and 2022 in the references.

Experimental design

The picture quality of the presented graphs is very poor. Use .emf format if possible to improve the quality of pictures.
The equations are also very dull. Use MathType software to write the equations.

Validity of the findings

The results of training loss, validation loss, training MAE, validation MAE, training MSE and validation MSE with respect to epochs are all missing. Include these graphs to show the actual and real performance of the models.

---

## Round 0.2 · Major Revisions

The revision requested by the referees has been made, but there are still issues that need to be corrected. It's essential to address the comments for a comprehensive revision

Reviewer 3 ·

Basic reporting

The author has effectively addressed the concerns raised in the previous review. The paper now demonstrates clear and professional English writing. It remains well-organized with appropriate inclusion of figures, tables, and raw data. The literature references still provide sufficient background and context for the research. The previous issues with typos and incorrect subsection numbering have been successfully rectified.

Experimental design

The author has successfully addressed the concerns raised in the previous review. Specifically, they have clarified the rationale for selecting specific data subsets from the Kaggle competition dataset and revised the figure and its description in Section 2.3 to eliminate any misleading aspects. Additionally, the explanation for the standard deviation error (SDE) definition has been provided in Section 2.9. The author has provided additional insights into the observed discrepancies in the performance of proposed and baseline models on the training and test sets for a comprehensive understanding.

Validity of the findings

The author has provided further explanation and clarification regarding the rationale behind the chosen dataset subset, the definition of the standard deviation error (SDE), and the claim regarding simpler models' performance.

Additional comments

The author has effectively addressed the previous concerns related to language, organization, and presentation. Notably, typos and incorrect subsection numbering have been rectified. In terms of content, the author has appropriately refrained from claiming the proposal of the LSTM architecture in the abstract, stating instead that they proposed a method based on LSTM. Additionally, the terminology in Section IV has been corrected, using "data preprocessing" instead of "feature selection." Overall, the paper remains clear, well-structured, and professionally written.

Reviewer 5 ·

Basic reporting

It is written well.

Experimental design

The design is good.

Validity of the findings

The authors have analyzed the results carefully.

Additional comments

The authors have completed the suggestions of the reviewers.

Reviewer 6 ·

Basic reporting

On the second page, line 42, the first reference number is given as 29. The order of citation numbers should be reconsidered.

A graph of the data can be added to better understand the data.

The provided figures have low resolution and should be renewed.

The prediction error values of the studies in the literature (for the next hour made with other data) should also be added to the literature in the introduction.

Experimental design

More information should be given about why the wind power forecast for the next hour is made. How does this forecast benefit the Wind industry and the power grid?

Validity of the findings

In this study, four different methods were used and compared with each other. But is the forecast error found sufficient for a subsequent hourly wind forecast study? Looking at the MAE error values, it is understood that the wind power data is normalized in the range of 0-1. It can be seen that the MAE error of the test data varies between approximately 0.04 and 0.08. Considering that the power value is normalized, it seems that the error varies approximately between 4-8%. Is this error value range an appropriate value for the next hour wind power forecast? The way to understand this is to add the error values in other estimation studies using the same data to the result section to determine whether it is appropriate or not.

Additional comments

In this study, wind power forecast was made for the next hour. Wind power data available on the Kaggle website was used as data. Four different methods were used as methods. GPeANN, DNN-MRT, Stacked LSTM and Bidirectional LSTM methods were compared. MAE, SDE, RMSE were determined as error criteria.

On the second page, line 42, the first reference number is given as 29. The order of citation numbers should be reconsidered.

The provided figures have low resolution and should be renewed.

In this study, four different methods were used and compared with each other. But is the forecast error found sufficient for a subsequent hourly wind forecast study? Looking at the MAE error values, it is understood that the wind power data is normalized in the range of 0-1. It can be seen that the MAE error of the test data varies between approximately 0.04 and 0.08. Considering that the power value is normalized, it seems that the error varies approximately between 4-8%. Is this error value range an appropriate value for the next hour wind power forecast? The way to understand this is to add the error values in other estimation studies using the same data to the result section to determine whether it is appropriate or not.

More information should be given about why the wind power forecast for the next hour is made. How does this forecast benefit the Wind industry and the power grid?

The prediction error values of the studies in the literature (for the next hour made with other data) should also be added to the literature in the introduction.

---

## Round 0.3 · accepted · Accept

Thank you for submitting your manuscript to PeerJ Computer Science. After reviewing the revised paper, it is my judgment that your manuscript is accepted for publication.

Reviewer 6 ·

Basic reporting

The changes noted in the previous review have been made.

The order of citation numbers have been corrected.

The graphs of the data have been added to understand the data better.

The resolutions of the figures have been increased.

Experimental design

A justification for the wind power forecast for the next hour from the latest related work has been provided in the introduction section.

The benefits of the next hour wind forecasting have been added.

Validity of the findings

Some of the information that was requested to be added in the previous review about the prediction error has been added.